# The Extracts of *Polygonum cuspidatum* Root and Rhizome Block the Entry of SARS-CoV-2 Wild-Type and Omicron Pseudotyped Viruses via Inhibition of the S-Protein and 3CL Protease

**DOI:** 10.3390/molecules27123806

**Published:** 2022-06-13

**Authors:** Shengying Lin, Xiaoyang Wang, Roy Wai-Lun Tang, Hung Chun Lee, Ho Hin Chan, Sheyne S. A. Choi, Tina Ting-Xia Dong, Ka Wing Leung, Sarah E. Webb, Andrew L. Miller, Karl Wah-Keung Tsim

**Affiliations:** 1Division of Life Science and Centre for Chinese Medicine, The Hong Kong University of Science and Technology, Hong Kong, China; lishlin@ust.hk (S.L.); wangxiaoyang@ust.hk (X.W.); roytwl@ust.hk (R.W.-L.T.); leoleehc@ust.hk (H.C.L.); botina@ust.hk (T.T.-X.D.); lkwing@ust.hk (K.W.L.); 2State Key Laboratory of Molecular Neuroscience, The Hong Kong University of Science and Technology, Hong Kong, China; hhbchan@hku.hk (H.H.C.); ssachoi@ust.hk (S.S.A.C.); barnie@ust.hk (S.E.W.); almiller@ust.hk (A.L.M.)

**Keywords:** SARS-CoV-2, omicron variant, traditional Chinese medicine, pseudovirus entry, Polygoni Cuspidati Rhizoma et Radix, zebrafish larvae

## Abstract

COVID-19, resulting from infection by the SARS-CoV-2 virus, caused a contagious pandemic. Even with the current vaccines, there is still an urgent need to develop effective pharmacological treatments against this deadly disease. Here, we show that the water and ethanol extracts of the root and rhizome of *Polygonum cuspidatum* (Polygoni Cuspidati Rhizoma et Radix), a common Chinese herbal medicine, blocked the entry of wild-type and the omicron variant of the SARS-CoV-2 pseudotyped virus into fibroblasts or zebrafish larvae, with IC_50_ values ranging from 0.015 to 0.04 mg/mL. The extracts were shown to inhibit various aspects of the pseudovirus entry, including the interaction between the spike protein (S-protein) and the angiotensin-converting enzyme II (ACE2) receptor, and the 3CL protease activity. Out of the chemical compounds tested in this report, gallic acid, a phytochemical in *P. cuspidatum*, was shown to have a significant anti-viral effect. Therefore, this might be responsible, at least in part, for the anti-viral efficacy of the herbal extract. Together, our data suggest that the extracts of *P. cuspidatum* inhibit the entry of wild-type and the omicron variant of SARS-CoV-2, and so they could be considered as potent treatments against COVID-19.

## 1. Introduction

Since its outbreak in December 2019, COVID-19, which resulted from variants of the SARS-CoV-2 virus, has infected and killed millions of people globally. Indeed, as of January 2022, over 520 million confirmed cases have been recorded, and the death toll has reached >6 million [1]. Numerous efforts have been made to develop effective therapeutics against this disease, and several treatments have been shown to prevent/reduce the risk of contracting the virus, including the oral drugs Paxlovid and Lagevrio, as well as antibody treatments [2]. Various SARS-CoV-2 mutations have been generated so far; these were given Greek letter monikers, from alpha to omicron. Each variant has distinct characteristics; for example, omicron is more highly infective than wild-type virus and has been found to reduce the effectiveness of current treatments, including the vaccines being employed today [3]. Therefore, there is an urgent need to discover new efficacious inhibitors to protect and/or treat this deadly disease, in order to complement the treatments that are already in use.

Viral infection is initiated when viral particles bind to host cells via fundamental receptors. The mechanism of viral entry and the replication of SARS-CoV-2 has been well documented, and it is known that although endosomal and non-endosomal pathways are both routes of viral entry, the endosomal route appears to be the more common of the two [4]. Several key proteins play crucial roles in the endosomal pathway. These include the spike (S)-protein and 3CL protease, expressed by the virus, and angiotensin-converting enzyme II (ACE2) expressed by the host cells. The S-protein comprises two subunits, S1 and S2. The S1 subunit recognizes the ACE2 receptor and forms an ACE2/S-protein complex on the surface of the host cells, whereas the S2 subunit triggers cell fusion, which allows the virus to pass through the cell membrane and activates a signaling cascade [5]. Subsequently, viral replication is mediated by a group of non-structural proteins, such as 3CL protease, which activates the downstream signaling events that regulate viral replication and ultimately lead to somatic damage of the host cells. These proteins form a signaling network that drives the entry and replication of the virus. Therefore, they are potential targets for developing potent treatments to tackle this disease [6].

Chinese herbal medicines have been utilized for thousands of years, and in that time, they have made significant contributions to combating various viral diseases. For example, herbal medicines were shown to exhibit a remarkable efficacy against SARS-CoV-1 (which arose in China in 2003), by relieving symptoms and shortening the course of the disease. Most recently, several herbal prescriptions, including *Jinhua Qinggan* granules and *Lianhua Qingwen* capsules, have been recommended for patients suffering from COVID-19 in China [7,8]. In light of this, it is strongly believed that Chinese medicine is capable of making a contribution to combating SARS-CoV-2 [7,8].

To search for additional effective inhibitor(s) of SARS-CoV-2 infection, we established a drug screening platform that tests the efficacy of different herbal extracts on various aspects of the viral entry pathway. To date, over 1000 herbal extracts/phytochemicals have been tested, and several have been shown to have an inhibitory effect [9]. As a result of confidentiality issues, these data are not described here. Among these hits, the root and rhizome of *Polygonum cuspidatum* Sieb. et Zucc (Polygoni Cuspidati Rhizoma et Radix; PCRR), a Chinese medicinal herb that has been used for many years, displayed robust inhibitory activities against the entry of wild-type and the omicron variant of the SARS-CoV-2 pseudotyped virus. We demonstrated that the water and ethanol extracts of PCRR, and one of its chemical constituents, gallic acid, significantly inhibited entry of the SARS-CoV-2 pseudovirus, largely by blocking the interaction of the ACE2/S-protein and inhibition of 3CL protease.

## 2. Results

When the HKCMMS (Hong Kong Chinese Materia Medica Standards) conducted an HPLC quality control of the water and ethanol extracts of PCRR; polydatin and resveratrol were both identified as chemical markers, while gallic acid was another potentially active chemical component. Therefore, these three chemicals were selected as chemical markers of PCRR in the subsequent experiments. In 100 g of the dried water extract (PCRR_water_), there was 1.24% polydatin, 0.45% resveratrol, and 0.14% gallic acid, whereas in the same amount of the ethanol extract (PCRR_EtOH_), there was 4.08% polydatin, 1.86% resveratrol, and 0.56% gallic acid (Figure 1). The extraction efficacy was ~28% and ~22% for the ethanol and water extracts, respectively, and, in general, PCRR_EtOH_ contained higher amounts of the main chemical components than PCRR_water_.

An ELISA assay was employed to determine the binding activities of the S-protein with ACE2. The aim of this series of experiments was to identify whether any of the herbal extracts or phytochemicals examined here inhibited this binding. A standard inhibitor (NIBSC code 20/136) acted as a positive control, as it demonstrated an ~80% inhibition in S-protein/ACE2 binding (Figure 2A). Our data showed that when compared with this standard inhibitor, PCRR_water_ and PCRR_EtOH_ displayed robust binding inhibition in a dose-dependent manner, ranging from 0.001 to 0.1 mg/mL. Both extracts had a very similar efficacy, with IC_50_ values of 0.01 mg/mL and 0.02 mg/mL for PCRR_water_ and PCRR_EtOH_, respectively. Moreover, when used at ~0.06 mg/mL, both extracts were potent inhibitors, inhibiting the ACE2/S-protein interaction by >80% (Figure 2B).

We also tested the ability of the PCRR herbal extracts to inhibit 3CL protease. A known inhibitor, GC376, served as a positive control, as it was shown to elicit ~80% inhibition of the enzyme. We showed that PCRR_EtOH_ displayed a good inhibitory activity in a dose-dependent manner from 0.01 to 1 mg/mL, and the extract exhibited a maximal inhibition of ~80% at a concentration of 0.25 mg/mL (Figure 2C). Similar to PCRR_EtOH_, PCRR_water_ also displayed dose-dependent inhibition, with a maximal inhibition of ~80% at 0.5 mg/mL. The efficacy of PCRR_EtOH_ was better than that of PCRR_water_, with an IC_50_ value of 0.1 mg/mL compared with 0.45 mg/mL (Figure 2C).

As parental HEK293T cells exhibit negligible expression of the ACE2 receptor, this cell line was transfected with cDNA encoding human *ACE2*, in order to establish a robust pseudovirus entry cell model to evaluate the activity of the PCRR extracts [9,10]. To determine the transfection efficiency of *ACE2* cDNA in the cultures, cells were immunolabeled with an anti-ACE2 antibody. The cells were also counterstained with DAPI to label the nucleus: the cytosol and plasma membrane were stained via the expression of ATPase. We showed that ~50% of the HEK293T-ACE2 cells over-expressed ACE2 (Figure 3), indicating that this cell-receptor model could be employed to test pseudovirus entry.

We tested the viral entry of wild-type or the omicron pseudovirus (which both expressed the S-protein and luciferase) into the host cells, and the amount of luciferase activity indicated the presence of the pseudovirus in the cells [11]. Initially, the viability of HEK293T cells following treatment with PCRR_EtOH_ or PCRR_water_ was investigated. Although a relatively low cell viability was observed at high concentrations of PCRR_EtOH_, the number of cells remained relatively constant at concentrations ranging between 0.001 and 0.025 mg/mL. In contrast, no significant apoptosis was detected following the treatment of PCRR_water_ at any concentration (Appendix A).

Next, we incubated HEK293T-ACE2 cells with varying amounts of wild-type pseudovirus. As expected, there was a gradual reduction in luciferase activity with the dilution of the pseudovirus (Figure 4A). A neutralizing antibody was utilized as a positive control, and this exhibited ~70% inhibition of pseudovirus entry. PCRR_water_ and PCRR_EtOH_ were both found to inhibit the viral entry of wild-type pseudovirus in dose-dependent manners, with corresponding IC_50_ values of 0.025 and 0.015 mg/mL, respectively (Figure 4B). Furthermore, both extracts also blocked entry of the omicron pseudovirus into the host cells, with a higher level of inhibition being achieved at high concentrations (0.05–0.1 mg/mL) for both PCRR extracts, when compared with those required for wild-type pseudovirus (Figure 4C). For both wild-type and the omicron pseudoviruses, PCRR_EtOH_ showed a better inhibition than PCRR_water_.

In addition to measuring the luciferase activity, we also measured the level of green fluorescence (derived from the *ZsGreen* plasmid expressed by wild-type pseudovirus), which was generated after the pseudovirus entered HEK293T-ACE2 cells. The highest intensity of green fluorescence was detected in the untreated control cells without any drug treatment, i.e., indicating the highest amount of viral entry, and no obvious green fluorescence was observed in the presence of the neutralizing antibody (Figure 5). This viral entry platform was then subjected to drug treatments with the PCRR extracts. We found that the fluorescence intensity was remarkably reduced when the cultures were incubated with PCRR_water_ (0.1 mg/mL) or PCRR_EtOH_ (0.1 mg/mL). These data further indicate that the PCRR extracts can inhibit the entry of the pseudovirus into the host cells and thus prevent host cell infection.

The effect of PCRR_water_ and PCRR_EtOH_ on wild-type pseudovirus entry into intact zebrafish larvae was also explored. In these experiments, the expression of the *luc* gene was used to indicate successful pseudovirus entry. As shown by the representative gel images (Figure 6A,B, left panels), PCRR_water_ and PCRR_EtOH_ both inhibited the expression of *luc*, in contrast with the untreated or DMSO-treated controls. The quantification of these data (Figure 6A,B, right panels) showed that the relative level of the *luc/g6pd* gene expression in both treatment groups was significantly lower for PCRR_water_ (at *p* < 0.05) and PCRR_EtOH_ (at *p* < 0.001), compared with their respective controls. Once again, the anti-viral effect of PCRR_EtOH_ was better than that of PCRR_water_. These results suggested that treatments with the water and ethanol extracts of PCRR can reduce pseudovirus entry in an intact animal model.

According to previous reports, the water and EtOH extracts of PCRR contain various phytochemicals, of which nine compounds (i.e., gallic acid, polydatin, emodin, physcion, anthraglycoside B, chrysophonal, rhein, resveratrol, and aloe emodin) were selected for subsequent anti-viral testing [12,13]. The highest tested concentrations of samples depended on their optimal solubility in the DMSO solvent and assay buffer. Interestingly, gallic acid displayed significant inhibition of viral entry into HEK293T cells in a dose-dependent manner, with an IC_50_ value of 23.5 µM (Figure 7A). In contrast, no significant inhibitory activities were observed from the other eight chemicals, indicating that they are not potent inhibitors of SARS-CoV-2 (Figure 7B). To validate our findings, a docking analysis of these phytochemicals against the S-protein was conducted. The receptor binding domain (RBD) between the S-protein and ACE2 receptor was identified, and this was selected for the subsequent computational docking study. K22 is a known inhibitor of the SARS-CoV-1 S-protein [5], and so this was utilized as a positive control. It was shown to have a binding energy of −12 KJ/mol. Consistent with our anti-viral tests described above, gallic acid was shown to bind to RBD the most efficiently, with a binding energy of −6.1 KJ/mol, whereas the others required a considerably higher binding energy (i.e., >−5 KJ/mol) (Appendix A). A docking analysis was also conducted between these phytochemicals and the S-protein of the omicron variant. Again, gallic acid was predicted to bind the most efficiently to the RBD, with a binding energy of −6.3 KJ/mol (Appendix A). These data suggest that it is the gallic acid component of the PCRR extracts that is responsible, at least in part, for inhibiting pseudoviral entry.

## 3. Discussion

Herbal medicine has been employed in fighting against pandemic incidents throughout history in China. This has resulted in the accumulation of many remarkable hypotheses and clinical methodologies for combating various diseases. According to the theory of Chinese medicine, COVID-19 is related to three initiating factors—dampness, heat, and a toxic pathogen [14]. To eliminate these factors from patients suffering from SARS-CoV-2 infection, several Chinese herbal prescriptions have been recommended by the National Health Commission of China and the World Health Organisation (WHO). These include *Huashi Baidu* formula, *Qingying* decoction, and *Xuanfei Baidu* formula; it is interesting to note that PCRR is one of the key herbs in the latter [14,15]. Not only does this imply that herbal extracts/phytochemicals might be employed in fighting against coronavirus, but it also indicates that PCRR itself could be especially effective against SARS-CoV-2 infection and so it might prove to be a key component of novel drugs developed for the treatment of COVID-19.

*Polygonum* plants have been broadly utilized in the food and pharmaceutical industry for many years. Recently, a few members of the *Polygonum* family have been found to demonstrate anti-viral activities. For example, an extract prepared from *P. perfoliatum* was reported to show activities against herpes simplex virus-1, the hepatitis B virus, and the influenza virus, by increasing the production of IgA and IgG antibodies [16]. In addition, PCRR and its phytochemical constituents, emodin and resveratrol, were reported to inhibit entry and replication of the influenza A virus [17]. Furthermore, gallic acid, one of the major constituents of PCRR, was predicted through computational analysis to have excellent inhibitory properties against SARS-CoV-2 infection [18]. These findings suggest that gallic acid along with its parental Chinese medicinal herbs are likely to exhibit anti-viral effects against SARS-CoV-2. Indeed, our new data show that PCRR_EtOH,_ PCRR_water_ and gallic acid all exhibited robust inhibition to the entry of the SARS-CoV-2 pseudovirus. As the amount of gallic acid in PCRR_EtOH_ and PCRR_water_ was shown to be 0.56% and 0.14% per dried weight, respectively, this might explain why PCRR_EtOH_ was more effective than PCRR_water_ in our testing platforms. Nevertheless, gallic acid is just one component of PCRR, and so other (as yet unidentified) phytochemicals might play a role in the anti-SARS-CoV-2 activity of these herbal extracts. Indeed, Nawrot-Hadzik et al. [19] reported that several compounds from the PCRR herb, including vanicoside A and vanicoside B, can inhibit the main protease (Mpro), with IC_50_ values of 23.10 µM and 43.59 µM, respectively, which indicate that both compounds might account for the anti-COVID-19 effectiveness of PCRR. Furthermore, extensive investigations were conducted previously and revealed that PCRR consisted of more than 100 phytochemicals [13,20,21]. This implied that a broad screening of these chemicals would enable us to further identify other PCRR phytochemicals containing anti-viral properties. Besides, a more in-depth HPLC analysis utilizing a higher concentration of a polar solvent such as acetonitrile might discover other crucial compounds.

In summary, to date, various phytochemicals with anti-viral properties against SARS-CoV-2 have already been identified. For example, theaflavin (a flavonoid derived from black tea) was shown to inhibit SARS-CoV-2 by interrupting the RNA polymerase, and lycorine (from *Lycoris radiata*) was shown to inhibit the growth of the virus with an IC_50_ value of 15.7 nM [22]. Furthermore, several alkaloids derived from *Stephania tetrandra* (i.e., tetrandrine, fangchinoline, and cepharanthine) have shown promising anti-SARS-CoV-2 effects [22]. Moreover, epigallocatechin gallate (EGCG; from green tea and the root of *Polygonum multiflorum*) was shown to inhibit the S-protein/ACE2 interaction and demonstrate robust attenuation of SARS-CoV-2 infection with IC_50_ values ranging from 25 to 500 µg/mL [9]. Interestingly, both EGCG and gallic acid share a gallate scaffold. This suggests that other phytochemicals with this chemical scaffold might also display significant binding to the SARS-CoV-2 S-protein and thus exhibit considerable anti-viral properties.

## 4. Materials and Methods

### 4.1. Cell Culture

HEK293T cells (American Type Culture Collection, Manassas, VA, USA) were maintained in high glucose Dulbecco’s Modified Eagle Medium (DMEM) containing 1% penicillin/streptomycin and 10% fetal bovine serum (Thermo Fisher Scientific, Waltham, MA, USA; herein called culture medium) at 37 °C in an incubator under a water-saturated atmosphere and 5% CO_2_. The culture medium was freshly provided every other day. HEK293T cells overexpressing human ACE2 (hACE2) were prepared by transfection with the pcDNA3.1-hACE2 plasmid (Addgene, Watertown, MA, USA). The cell viability was detected following methodology as described previously [23], except the absorbance here was measured at 490 nm.

### 4.2. Preparation and HPLC Analysis of Herbal Extracts

The herb PCRR was purchased from the local herbal market and was authenticated in accordance with HKCMMS [24]. PCRR powder (10 g) was placed in a 250 mL round-bottomed flask and dissolved in 100 mL distilled water or 90% ethanol to obtain the water and ethanol extracts, respectively. The solution was then refluxed for 1 h prior to being filtered through a paper filter (110 µm, Advantec, Tokyo, Japan). The extracts were then evaporated to dryness utilizing a rotary evaporator to provide yields of 2.80 g ethanol extract (hereafter called PCRR_EtOH_) or 2.18 g water extract (hereafter called PCRR_water_). HPLC was performed following the methodology described by HKCMMS [24], utilizing PCRR_water_ (0.1 mg/mL), PCRR_EtOH_ (0.1 mg/mL), polydatin (0.04 mg/mL), resveratrol (0.04 mg/mL), and gallic acid (0.04 mg/mL). The gradient was as follows: 0–12 min with 80% water, 20% acetonitrile; 12–13 min with 80–70% water, 20–30% acetonitrile; and 13–30 min with 70% water, 30% acetonitrile. A detection wavelength of 306 nm was used.

### 4.3. Production of the SARS-CoV-2 Pseudotyped-Virus

HEK293T cells placed at 80% confluence were transfected with various components of the SARS-related coronavirus 2, Wuhan-Hu-1 Spike-pseudotyped Lentiviral Kit (NR-52948; BEI Resources, National Institute of Allergy and Infectious Diseases, Rockville, MD, USA), including the SARS-CoV-2 spike glycoprotein (NR-52514 for wild-type or 179907 for the omicron variant), a lentiviral backbone expressing gene *Luciferase* and *ZsGreen* (NR-52516), and several helper plasmids (NR-52517, NR-52518, and NR-52519) using Lipofectamine™ 3000 (Thermo Fisher Scientific) or JetPRIME (Polyplus, Shanghai, China) transfection reagent, following instructions from the manufacturers. After 72 h, the particles of the SARS-CoV-2 pseudotyped-virus (defined as pseudovirus) were collected and passed through a 0.45 µm filter (Sartorius, Germany) before being used directly in most experiments. However, for the zebrafish test, wild-type pseudovirus was purified utilizing polyethylene glycol (PEG; hereafter called PEG-pseudovirus) [25], and the culture medium was replaced with phosphate buffered saline (PBS; 137 mM NaCl, 2.7 mM KCl, 10 mM Na_2_HPO_4_, 1.8 mM KH_2_PO_4_ pH 7.4). Both the pseudovirus and PEG-pseudovirus were stored at −80 °C until further required.

### 4.4. Entry of the SARS-CoV-2 Pseudovirus

HEK293T cells overexpressing the ACE2 receptor were seeded onto 48-well plates and were incubated with 400 µL culture medium containing pseudovirus (either wild-type or the omicron variant, 100 µL) and either one of the PCRR extracts (either PCRR_water_ or PCRR_EtOH_ at various concentrations) or gallic acid at 37 °C for 24 h. This medium was subsequently replaced with fresh medium, and the cultures were allowed to recover for 48 h, after which they were washed with PBS before conducting the luciferase assay. PCRR_EtOH_ or PCRR_water_ were tested at final concentrations of 0.001 to 0.1 mg/mL, whereas gallic acid was tested at 1 to 100 µM. An anti-SARS-CoV-2 neutralizing antibody (A19215, ABClonal, Woburn, MA, USA) served as the positive control (1 µg/mL), while a solvent blank without the pseudovirus was employed as the negative control. The amount of inhibition was detected according to the luciferase activity with extracts/phytochemicals treatments and was normalized to the luciferase activity without any treatments.

To determine viral entry in the zebrafish model, larvae at 3 days post-fertilization (dpf) were treated for 6 h with Danieau’s solution (17.4 mM NaCl, 0.21 mM KCl, 0.12 mM MgSO_4_·7H_2_O, 0.18 mM Ca (NO_3_)_2_·4H_2_O, and 1.5 mM HEPES; pH 7.2) ±0.125 mg/mL PCRR_water_, or with Danieau’s solution containing 0.1% DMSO ± 0.030 mg/mL PCRR_EtOH_ in 24-well plates (SPL Life Sciences, Gyeonggi-do, Korea). The experiment was performed using 10 embryos in each group (*n* = 4). The larvae were subsequently transferred onto 96-well plates (SPL Life Sciences), one larva per well, containing the respective treatment solution and 8 µL PEG-pseudovirus. The larvae were subsequently incubated at 28 °C for 72 h, until they reached ~6.25 dpf. At this stage, the larvae from each treatment group were pooled and washed with Milli-Q water for 6 × 20 min with gentle agitation before RNA extraction.

### 4.5. Immunofluorescence of Protein Staining

The cultured cells were fixed with 4% paraformaldehyde for 30 min. Samples were incubated with 5% BSA for 1 h. For the transfection efficiency of *ACE2*, an immunofluorescence assay was performed, as previously described [26], using an anti-ATPase antibody (EP1845Y, Abcam, Cambridge, UK), a plasma membrane loading control (Abcam), and a primary ACE2 antibody (E-11, Santa Cruz Biotechnology, Santa Cruz, CA, USA) all at a dilution of 1:400, at 4 °C overnight, followed by incubation with Alexa 647 and 488 conjugated antibodies (Abcam Ltd., Cambridge, UK), respectively. After immunolabeling, the samples were mounted with ProLong^®^ Gold Antifade Mountant containing DAPI (Cell Signaling Technology, Danvers, MA, USA). For pseudovirus entry, the infected cells were obtained following the method from 4.4. Zsgreen was introduced to the pseudovirus beforehand, while the F-actin component of the cell membrane was stained with fluorescent phalloidin. All of the images were visualized and obtained utilizing a confocal inverted laser microscope (Leica SP8) with 48× magnification. The emission wavelength was 405 nm for DAPI, 488 nm for Zsgreen, and 505 nm for F-actin/phalloidin.

### 4.6. Luciferase Assay

A luciferase assay was performed following the methodology, as previously described [9,23]. The inhibition in percentage of every sample was determined as follows: Inhibition rate = (luciferase activity of the solvent blank − luciferase activity of the sample)/(luciferase activity of the solvent blank − luciferase activity of group without pseudovirus) × 100%.

### 4.7. Inhibition of Spike Protein

The inhibition of the spike protein was calculated using a SARS-CoV-2 Spike-ACE2 binding assay kit (ImmunoDiagnostics, Hong Kong, China), following instructions from the manufacturer. A standard inhibitor (calibrated to NIBSC code 20/136), provided by the manufacturer, was employed as the positive control. The reaction was completed by adding 2 M H_2_SO_4_ solvent and the data were quantified using a microplate reader (FlexStation; Molecular Devices, San Jose, CA, USA). The percentage inhibition was determined as follows: percentage inhibition = (P_Avg_ − S_Avg_)/P_Avg_ × 100%, where P_Avg_ and S_Avg_ are the mean OD values of the positive control and tested samples, respectively.

### 4.8. Inhibition of the 3CL Protease

The anti-3CL protease activities of the extracts were detected on a fluorogenic substrate with the SensoLyte SARS-CoV-2 3CL Protease Activity assay kit (BPS Bioscience, San Diego, CA, USA), following the instructions from the manufacturer. When 3CL protease was bound with the substrate, the wavelengths of the excitation and emission fluorescence were measured at 360 nm and 460 nm, respectively. The percentage inhibition was calculated as follows: percentage inhibition = (P_Avg_, b − S_Avg_, b)/P_Avg_, b × 100%; where P_Avg_, b and S_Avg_, b represent the mean fluorescence of the positive control and test sample, respectively, subtracted from the mean fluorescence of the blank.

### 4.9. RNA Extraction and RT-PCR

RNA extraction and reverse transcription were conducted, as previously described [27]. The resulting cDNA generated was then amplified by PCR using the 2x Rapid Taq Master Mix (Vazyme Biotech, Nanjing, China), following instructions from the manufacturer. The PCR products were isolated using a 2% agarose gel, and the band intensities were quantified utilizing ImageJ. The level of luciferase (*luc*) mRNA was detected against that of glucose-6-phosphate dehydrogenase (*g6pd*) mRNA. The expression levels following treatment with PCRR_water_ or PCRR_EtOH_ were measured relative to their respective controls. The primers used were as follows: glucose-6-phosphate dehydrogenase -Fwd: 5′ TGC TTC CAC CAG CTC TGA TG 3′ and Rev: 5′ CCC TCA ACT CAT CAC TGC GT 3′; luciferase-Fwd: 5′ AAA CGC TTC CAC CTA CCA GG 3′ and Rev: 5′ TCC ACG ATC TCC TTC TCG GT 3′.

### 4.10. Computational Docking Studies

The structure of S-protein was downloaded from the Protein Data Bank (https://www.rcsb.org/ accessed on 1 May 2022), while the chemical structures of the phytochemicals were downloaded from Pubchem (https://pubchem.ncbi.nlm.nih.gov/ accessed on 1 May 2022). Virtual screening was carried out using software SEESAR (Version 11.0, https://www.biosolveit.de/ accessed on 1 May 2022) as follows: (i) The binding site for docking was selected and determined according to the residues forming a druggable pocket. Ligand binding states including protonation and tautomeric forms were subsequently processed and evaluated through the ProToss method to generate the most accessible hydrogen network. (ii) Docking modulation was conducted utilizing the “Compute LeadIT Docking” mode in the FlexX algorithm; ten binding conformations for each ligand were generated. (iii) The binding energy (i.e., ∆G) and estimated HYDE affinity (KiHYDE) for each ligand pose were determined through the “Assess Affinity with HYDE in SEESAR” mode in the HYDE rescoring function [9,28].

## 5. Conclusions

As COVID-19 has caused millions of deaths and resulted in unprecedented damage to all aspects of our lives, highly effective treatments are urgently needed to tackle this deadly disease. Here, we described the simple and efficient screening platform we developed (comprising both *in vitro* and *in vivo* methodologies) to provide a first approach for testing the efficacy of different herbs against SARS-CoV-2 entry. As a result, the extracts of PCRR were found to display promising potency in inhibiting S-protein/ACE2 binding and 3CL protease activity. The herbal extracts also inhibited entry of wild-type SARS-CoV-2 pseudovirus into both HEK293T-ACE2 cells and zebrafish larvae, and of the omicron variant into HEK293T-ACE2 cells. Gallic acid, a component of PCRR, was shown to be responsible for at least some of the anti-viral effects of PCRR. Together, our data suggest that PCRR might be considered for use in clinical practices as one of potential treatments for COVID-19.

## Figures and Tables

**Figure 1 molecules-27-03806-f001:**
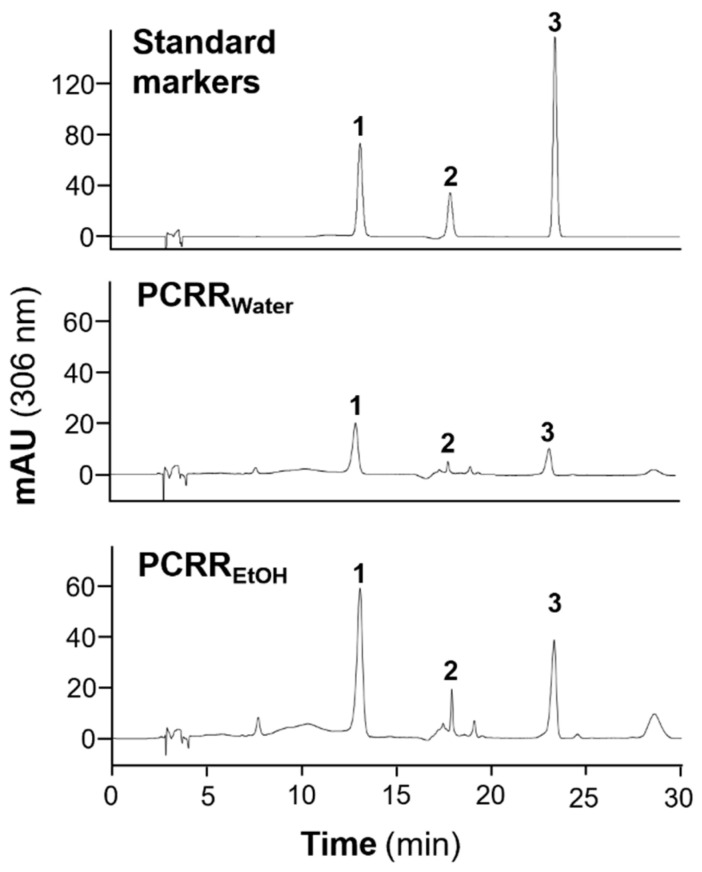
HPLC analysis of the PCRR extracts. The characteristic peaks of three standards, and the peaks obtained for PCRR_water_ and PCRR_EtOH_, were detected at an absorbance of 306 nm. The three peaks are (**1**) polydatin, (**2**) gallic acid, and (**3**) resveratrol.

**Figure 2 molecules-27-03806-f002:**
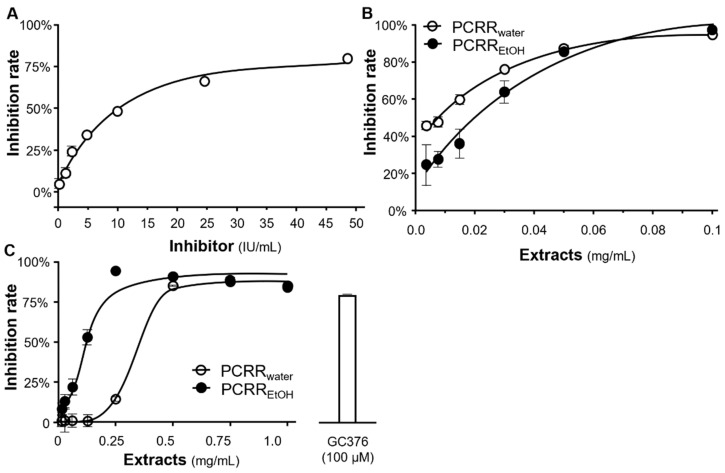
Effect of PCRR_water_ and PCRR_EtOH_ on the inhibition of S-protein-ACE2 binding and 3CL protease activity. (**A**) Line graph showing the inhibition rate of a standard inhibitor (calibrated to NIBSC code 20/136), which was employed as a positive control. (**B**,**C**) PCRR_water_ and PCRR_EtOH_ both inhibited, (**B**) S-protein-hACE2 binding, and (**C**) 3CL protease in dose-dependent manners. GC-376 served as a positive control. The inhibition rate was determined from the signal normalized to the control without the extracts. The data represent mean ± SD; *n* = 3.

**Figure 3 molecules-27-03806-f003:**
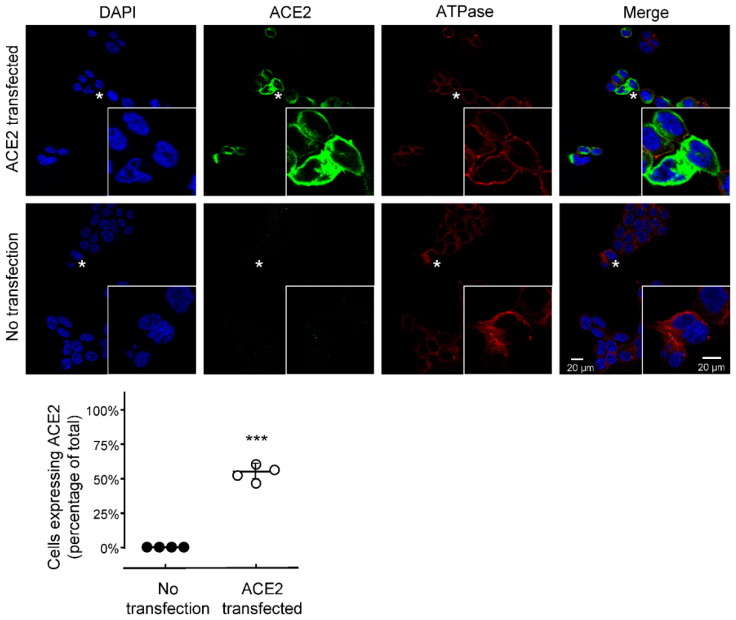
Over expression of ACE2 in HEK293T cells. Cultured HEK293T cells were transfected with cDNA encoding *ACE2* for 48 h before being immunolabeled with an anti-ACE2 antibody (in green). The cell nucleus was labelled with DAPI (in blue), while the cytosol and plasma membrane were stained via the expression of ATPase (in red). The cells were visualized, and the images were acquired with a Leica SP8 laser scanning confocal microscope. In each panel, the region indicated by the asterisk is shown in a higher magnification in the inset panel. When the data were quantified, the results showed that >50% of the HEK293T cells had successfully been transfected with *ACE2* cDNA. The data represent the mean ± SD; *n* = 4. The statistical differences were detected utilizing the two-sample *t*-test, *** *p* < 0.001.

**Figure 4 molecules-27-03806-f004:**
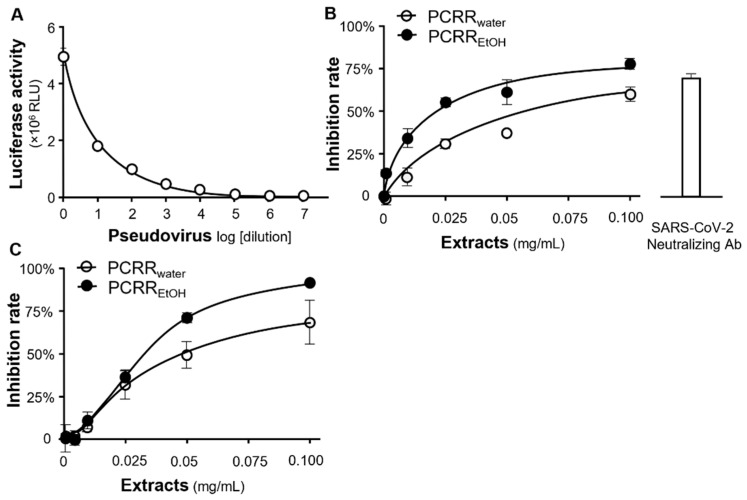
Inhibition of pseudovirus entry. (**A**) The intracellular luciferase activity (implying the rate of pseudovirus entry in RLU) was detected at different concentrations/dilutions of wild-type pseudovirus. (**B**) The entry of wild-type pseudovirus was disrupted by PCRR_water_ and PCRR_EtOH_ in dose-dependent manners. (**C**) These extracts also inhibited the entry of the omicron variant into the host cells. The inhibition rate was calculated based on the luciferase activity, which was normalized to the luciferase activity without any treatment. A SARS-CoV-2 neutralizing antibody at a concentration of 1 µg/mL was employed as a positive control. The data represent mean ± SD; *n* = 3.

**Figure 5 molecules-27-03806-f005:**
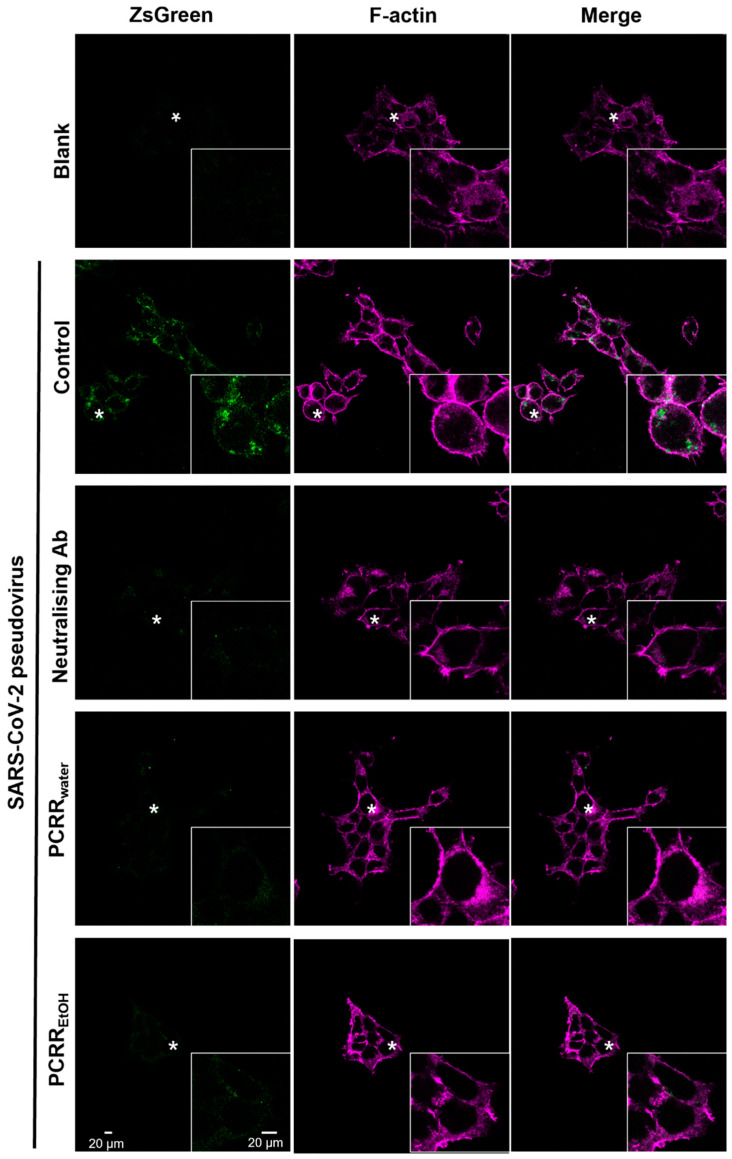
PCRR_EtOH_ and PCRR_water_ inhibit the entry of wild-type pseudovirus. The F-actin component of the cell membrane was stained with fluorescent phalloidin. The intensity of ZsGreen fluorescence in the cells was measured with ImageJ after treatment with PCRR_EtOH_ (0.1 mg/mL) or PCRR_water_ (0.1 mg/mL). The fluorescence intensity was considerably reduced when compared with the control group with no drug treatment. A SARS-CoV-2 neutralizing antibody (1 µg/mL) was utilized as a positive control. These are representative images; *n* = 3. In each image, the region marked with the asterisk is shown at a higher magnification in the inset panel.

**Figure 6 molecules-27-03806-f006:**
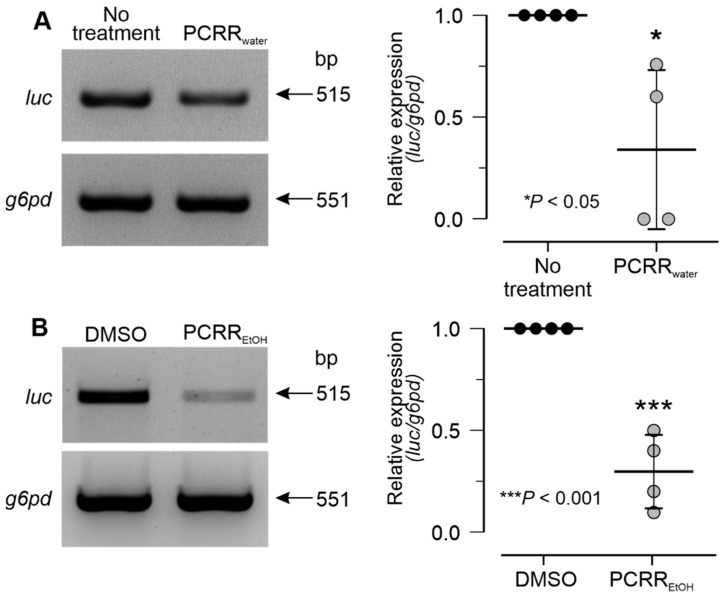
PCRR_water_ and PCRR_EtOH_ inhibit pseudovirus entry in the zebrafish test. Larvae at 3 dpf were pre-treated with (**A**) PCRR_water_ or (**B**) PCRR_EtOH_ before being co-treated with the respective PCRR extracts and the SARS-CoV-2 PEG-pseudovirus for 72 h. PCR amplification of whole larval cDNA indicated the relative level of *luc* expression in both the treatment and control groups. The relative level of *luc* expression was determined through normalization to the expression level of *g6pd*. The data represent the mean ± SD; (*n* = 4) and statistical differences were detected utilizing the two-sample *t*-test.

**Figure 7 molecules-27-03806-f007:**
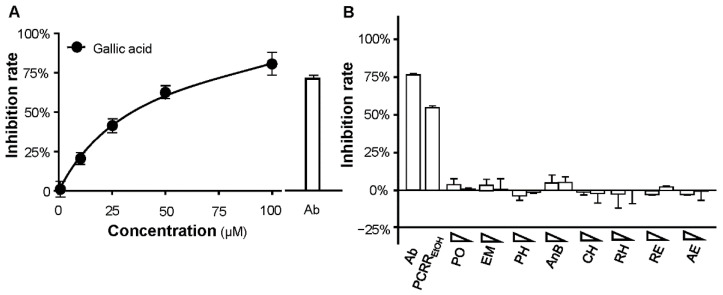
Gallic acid blocks entry of the pseudovirus into HEK293T cells. (**A**) Gallic acid displayed inhibition in a dose-dependent manner. (**B**) The other phytochemicals in PCRR showed a negligible inhibitory activity when tested at high and low concentrations. Ab: neutralizing antibody (used at 1 µg/mL); PCRR_EtOH_ (used at 0.025 mg/mL); PO: polydatin (used at 50 µM [high], 0.5 µM [low]); EM: emodin (used at 25 µM, 0.25 µM); PH: physcion (used at 1 µM, 0.01 µM); AnB: anthraglycoside B (used at 100 µM, 1 µM); CH: chrysophonal (used at 5 µM, 0.05 µM); RH: rhein (used at 20 µM, 0.2 µM); RE: resveratrol (used at 22 µM, 0.22 µM); AE: aloe emodin (used at 100 µM and 1 µM). All of the data are shown as mean ± SD; *n* = 3.

## Data Availability

The data presented in this study are available upon request from the corresponding author.

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
