# Peer review of "The Extracts of Polygonum cuspidatum Root and Rhizome Block the Entry of SARS-CoV-2 Wild-Type and Omicron Pseudotyped Viruses via Inhibition of the S-Protein and 3CL Protease"

_molecules, 2022, doi:10.3390/molecules27123806_

Round 1

Reviewer 1 Report

Check for typos, if any.

Author Response

  1. Comment 1: Check for typos, if any.

Our response: We have checked and revised the typos in the manuscript.

Reviewer 2 Report

The Authors showed that the water and ethanol extracts of the 14 root and rhizome of Polygonum cuspidatum (Polygoni Cuspidati Rhizoma et Radix), a common Chi- 15 nese herbal medicine, blocked the entry of wild-type and omicron variant of the SARS-CoV-2 16 pseudotyped virus into fibroblasts or zebrafish larvae. The extracts were shown to inhibit various aspects of the pseudovirus entry, including the 18 interaction between spike protein (S-protein) and angiotensin-converting enzyme II (ACE2), and 19 3CL protease activity. Although most of the phytochemicals within both the water and ethanol ex- 20 tracts had little effect on the pseudoviral entry, gallic acid, a phytochemical in P. cuspidatum, did 21 have a significant antiviral effect. Therefore, this might be responsible at least in part for the anti- 22 viral efficacy of the herbal extracts. The extracts of P. cuspidatum 23 inhibit the entry of the wild-type and omicron variants of SARS-CoV-2, they might be considered 24 for development as a first-line treatment for COVID-19.

The article has cientific value.

Author Response

  1. Comment 1: The article has scientific value.

Our response: Thank you.

Reviewer 3 Report

Dear Authors, I highly value the manuscript submitted for review-" The extracts of Polygonum cuspidatum root and rhizome block the entry of SARS-CoV-2 wild-type and omicron pseudotyped viruses via inhibition of the S-protein and 3CL protease” The work is interesting and provides important data. However, I noticed unacceptable mistakes and omissions of important facts that absolutely need to be corrected before publishing the article.

First, the introduction does not provide a sufficient background and contains too old references and outdated data, like:

·       Line 33, there's an old references, from 2020-“ Although numerous efforts have been made to develop effective treatments against this disease, there are currently still no therapeutics that provide 100% protection against the original SARS-CoV-2 virus, not to mention recent mutations of the virus that have emerged since the beginning of the pandemic [2]” It's not true that we don't have drugs. Please update this information. FDA approved antiviral drugs. See: https://www.fda.gov/consumers/consumer-updates/know-your-treatment-options-covid-19 You should mention that.

·       The introduction of this manuscript is too similar to your other article: Polygoni multiflori radix extracts inhibit SARS-CoV-2 pseudovirus entry in HEK293T cells and zebrafish larvae”

·       Line 56 - this is no longer " novel strategies "

·       Line 61 - Jinhua Qinggan granules and Lianhua Qingwen capsules. There is no citation to these recommendations

·       Line 63: You should change that sentence: “Thus, Chinese medicines have the potential to be developed as a first-line treatment against SARS-CoV-2 infection” , This is exaggerated. The evidence is not strong enough.

·       Line 72: gallic acid is not the major component according to the current literature. See article: https://doi.org/10.3390/molecules24061136

·       Line 67: your information from this line:” To date, over 1,500 herbal extracts/phytochemicals have been tested, and several have been shown to have an inhibitory effect [9].” In your article [9] you write about testing 1000 herbal extracts, not 1500. Moreover, you have not presented the results of these tests anywhere, neither in this manuscript nor in previous article - [9]. I propose to write in this manuscript that "this date is not presented"

Secondly, the information on the compounds contained in the rhizomes and roots of polygonum cuspidatum is obsolete, and does not contain new reports, also regarding the activity against 3 CL protease:

·       Line 203: “According to previous reports, there are nine major phytochemicals (i.e., gallic acid, polydatin, emodin, physcion, anthraglycoside B, chrysophonal, rhein, resveratrol and aloe emodin), in PCRR extracts [12-13], and so these chemicals were selected for subsequent antiviral testing.” 12 and 13 are old articles. You need to change the information, see: https://doi.org/10.3390/molecules24061136, DOI: 10.1055/a-0605-3857 . There are procyanidins and vanicosides - important active compounds also for inhibition of 3CL protease! Pharmaceuticals 2021, 14(8), 742; https://doi.org/10.3390/ph14080742 Read these articles and change you sentences and discussion, like in this line:

·       Line 259: “ Nevertheless, gallic acid is just one component of PCRR, and so other (as yet unidentified) phytochemicals might play a role in the anti-SARS-CoV-2 activity of these herbal extracts. Experiments to identify other active phytochemicals in PCRR are still required in the future investigation.”  It has already been done. See above articles! And here are also hplc-ms analysis: Pharmaceutics 2021, 13(11), 1764; https://doi.org/10.3390/pharmaceutics13111764

·       HPLC analysis: “The gradient was as follows: 0-12 min with 80% water, 20% acetonitrile; 12- 13 min with 80%-70% water, 20%-30% acetonitrile; and 13-30 min with 70% water, 30% 297 acetonitrile. A detection wavelength of 306 nm was used.”  This gradient allowed the identification of only a small fraction of compounds present in the extract. Washing the column with a higher concentration of acetonitrile would reveal other important compounds. You should add this information to your manuscript.

Other comments:

·       Line 20 in the abstract should be revised taking into account updated scientific literature. ” Although most of the phytochemicals within both the water and ethanol extracts had little effect on the pseudoviral entry, gallic acid, a phytochemical in P. cuspidatum, did have a significant antiviral effect. “The enzymes of viruses are influenced not only by gallic acid, but also by vanicosides and proanthocyanidins- Pharmaceuticals 2021, 14(8), 742; https://doi.org/10.3390/ph14080742

·       Line 23- “Together, our data suggest that as the extracts of P. cuspidatum  inhibit the entry of the wild-type and omicron variants of SARS-CoV-2, they might be considered for development as a first-line treatment for COVID-19. “ It is an exaggeration to write something like this, considering that we have some drugs for these viruses. In addition, clinical trials must first be conducted before the drug is introduced into treatment.

·       Line 77- please expand abbreviation HKCMMS

·       Line 207: IC50 of ~25 μM (Figure 7A).  IC 50 should be calculated accurately

·       Figue 7 b. why did you use different concentrations of test substances? “PO: polydatin (50 μM, 0.5 μM); 226 EM: emodin (25 μM, 0.25 μM); PH: physcion (1 μM, 0.01 μM); AnB: anthraglycoside B 227 (100 μM, 1 μM); CH: chrysophonal (5 μM, 0.05 μM); RH: rhein (20 μM, 0.2 μM); RE: 228 resveratrol (22 μM, 0.22 μM); and AE: aloe emodin (100 μM, 1 μM) “

·       Conclusion : you shouldn't write about developed screening platform and about tested- Out of 1,000 herbal extracts and over 200 phytochemicals because you didn't presented this results here.

In general, first of all, the manuscript should update the information about the SARS virus and its treatment, and about the compounds present in the rhizomes and roots of polygonum cuspidatum and their activity in relation to the 3CL protease.

Author Response

Reply to the reviewers#3

  1. Comment 1: Line 33, there's an old references, from 2020-“ Although numerous efforts have been made to develop effective treatments against this disease, there are currently still no therapeutics that provide 100% protection against the original SARS-CoV-2 virus, not to mention recent mutations of the virus that have emerged since the beginning of the pandemic [2]” It's not true that we don't have drugs. Please update this information. FDA approved antiviral drugs. See: https://www.fda.gov/consumers/consumer-updates/know-your-treatment-options-covid-19 You should mention that.

 Our response: We have updated the text and cited a new reference according to the latest publication. Please see line 32-35 in page 1.

  1. Comment 2: The introduction of this manuscript is too similar to your other article: “Polygoni multiflori radix extracts inhibit SARS-CoV-2 pseudovirus entry in HEK293T cells and zebrafish larvae”

 Our response: The two manuscripts were originated from the same authors, same project and particularly same scientific direction, hence we believe it is commonly acceptable that the two articles share a similar layout of introduction.

  1. Comment 3: Line 56 - this is no longer " novel strategies”. Line 61 - Jinhua Qinggan granules and Lianhua Qingwen capsules. There is no citation to these recommendations.

Our response: The mentioned wordings have been revised, please see line 56, page 2. The citation has been updated in line 63, page 2.

  1. Line 63: You should change that sentence: “Thus, Chinese medicines have the potential to be developed as a first-line treatment against SARS-CoV-2 infection” , This is exaggerated. The evidence is not strong enough.

Our response:  The wordings have been rephased in a non-exaggerated tone in line 63-64, page 2.

  1. Line 72: gallic acid is not the major component according to the current literature. See article: https://doi.org/10.3390/molecules24061136

Our response:  The mentioned wordings have been revised in line 74, page 2.

  1. Line 67: your information from this line:” To date, over 1,500 herbal extracts/phytochemicals have been tested, and several have been shown to have an inhibitory effect [9].” In your article [9] you write about testing 1000 herbal extracts, not 1500. Moreover, you have not presented the results of these tests anywhere, neither in this manuscript nor in previous article - [9]. I propose to write in this manuscript that "this date is not presented"

Our response: We initiated the first draft of our last paper [Ref 9] in December 2021, and we have made more progress since then; hence the number of screened herbal extracts/phytochemicals have been increased. To keep the consistency between the two reports, we have revised the number to “1,000” at this report in line 67, page 2. In addition, as we did not present our screening results in detail in this article, a description has been included in line 68-69, page 2.

  1. Line 203: “According to previous reports, there are nine major phytochemicals (i.e., gallic acid, polydatin, emodin, physcion, anthraglycoside B, chrysophonal, rhein, resveratrol and aloe emodin), in PCRR extracts [12-13], and so these chemicals were selected for subsequent antiviral testing.” 12 and 13 are old articles. You need to change the information, see: https://doi.org/10.3390/molecules24061136, DOI: 10.1055/a-0605-3857. There are procyanidins and vanicosides - important active compounds also for inhibition of 3CL protease! Pharmaceuticals 2021, 14(8), 742; https://doi.org/10.3390/ph14080742 Read these articles and change you sentences and discussion, like in this line.

Our response: We are fully aware that the cited references are relatively old (Qian et al. 2006 and Fu et al. 2015), as compared with the references you recommended (Nawrot-Hadzik et al., 2018 and Nawrot-Hadzik et al., 2019). However, we have to highlight that in the two new articles mentioned by you, acetone solvent was mainly utilised in the sample extraction and preparation. In general, acetone has excellent solubility in dissolving various chemicals, which therefore could account for a large number of chemicals being detected by Nawrot-Hadzik et al. in HPLC. In the reports of Qian et al. (2006) and Fu et al. (2015), MeOH and water were the major solvents in extraction, respectively. The solvents being used are very much in line with our sample preparations of PCRREtOH and PCRRwater. Besides, EtOH and water have been widely utilised for extraction by TCM practitioners and industries. Taken together, we believe that the reports of Qian et al. (2006) and Fu et al. (2015) can be employed as profound references/guidance in obtaining the chemical composition of PCRR. In view of your comment, we have modified the text in line 205-208, page 7.

  1. Line 259: “Nevertheless, gallic acid is just one component of PCRR, and so other (as yet unidentified) phytochemicals might play a role in the anti-SARS-CoV-2 activity of these herbal extracts. Experiments to identify other active phytochemicals in PCRR are still required in the future investigation.” It has already been done. See above articles! And here are also hplc-ms analysis: Pharmaceutics 2021, 13(11), 1764; https://doi.org/10.3390/pharmaceutics13111764

Our response: We have taken the report of Nawrot-Hadzik et al. (2021) into account and revised the text in the discussion, line 266-270, page 9.

  1. HPLC analysis: “The gradient was as follows: 0-12 min with 80% water, 20% acetonitrile; 12- 13 min with 80%-70% water, 20%-30% acetonitrile; and 13-30 min with 70% water, 30% 297 acetonitrile. A detection wavelength of 306 nm was used.” This gradient allowed the identification of only a small fraction of compounds present in the extract. Washing the column with a higher concentration of acetonitrile would reveal other important compounds. You should add this information to your manuscript.

Our response: The HPLC analysis in this report was performed strictly following the methodology developed by HKCMMS, including selection of chemical markers (plus a potential active ingredient--gallic acid in our case) and development of eluent gradient. The aim of this analysis was to ensure the authenticity of TCM herb, as well as its corresponding extracts meeting the requirement of HKCMMS. On the other hand, we are fully aware that “washing the column with a higher concentration of acetonitrile would reveal other important compounds”; hence we have included this part of description in the discussion, line 270-272, page 9.

Other comments

  1. Line 20 in the abstract should be revised taking into account updated scientific literature. ” Although most of the phytochemicals within both the water and ethanol extracts had little effect on the pseudoviral entry, gallic acid, a phytochemical in P. cuspidatum, did have a significant antiviral effect. “The enzymes of viruses are influenced not only by gallic acid, but also by vanicosides and proanthocyanidins-Pharmaceuticals 2021, 14(8),742; https://doi.org/10.3390/ph14080742

Our response: We have taken the latest scientific literatures into account and revised the text in line 20-22, page 1.

  1. Line 23- “Together, our data suggest that as the extracts of P. cuspidatum inhibit the entry of the wild-type and omicron variants of SARS-CoV-2, they might be considered for development as a first-line treatment for COVID-19. “ It is an exaggeration to write something like this, considering that we have some drugs for these viruses. In addition, clinical trials must first be conducted before the drug is introduced into treatment.

Our response: We have modified the text as line 22-24, page 1.

  1. Line 77- please expand abbreviation HKCMMS.

Our response: We have given the full name and abbreviation of HKCMMS in line 78, page 2.

  1. Line 207: IC50 of ~25 μM (Figure 7A). IC 50 should be calculated accurately

Our response: We have calculated accurately and revised the IC50 value in line 211, page 8.

  1. Figue 7 b. why did you use different concentrations of test substances? “PO: polydatin (50 μM, 0.5 μM); 226 EM: emodin (25 μM, 0.25 μM); PH: physcion (1 μM, 0.01 μM); AnB: anthraglycoside B 227 (100 μM, 1 μM); CH: chrysophonal (5 μM, 0.05 μM); RH: rhein (20 μM, 0.2 μM); RE: 228 resveratrol (22 μM, 0.22 μM); and AE: aloe emodin (100 μM, 1 μM) “

Our response: For a compound in the test, the highest concentration being tested was depending on its optimal solubility in DMSO and assay buffer. The samples in low concentration were simply obtained by diluting the sample at 100 times. We have revised the wordings in line 208-209, page 8.

  1. Conclusion: you shouldn't write about developed screening platform and about tested- Out of 1,000 herbal extracts and over 200 phytochemicals because you didn't presented this results here.

Our response: We have revised the conclusion section in line 420-422, page 12. 

Round 2

Reviewer 3 Report

Line 203. I do not agree with one answer of the authors: “

  1. Line 203: “According to previous reports, there are nine major phytochemicals (i.e., gallic acid, polydatin, emodin, physcion, anthraglycoside B, chrysophonal, rhein, resveratrol and aloe emodin), in PCRR extracts [12-13], and so these chemicals were selected for subsequent antiviral testing.” 12 and 13 are old articles. You need to change the information, see: https://doi.org/10.3390/molecules24061136, DOI: 10.1055/a-0605-3857. There are procyanidins and vanicosides - important active compounds also for inhibition of 3CL protease! Pharmaceuticals 2021, 14(8), 742; https://doi.org/10.3390/ph14080742 Read these articles and change you sentences and discussion, like in this line.

Our response: We are fully aware that the cited references are relatively old (Qian et al. 2006 and Fu et al. 2015), as compared with the references you recommended (Nawrot-Hadzik et al., 2018 and Nawrot-Hadzik et al., 2019). However, we have to highlight that in the two new articles mentioned by you, acetone solvent was mainly utilised in the sample extraction and preparation. In general, acetone has excellent solubility in dissolving various chemicals, which therefore could account for a large number of chemicals being detected by Nawrot-Hadzik et al. in HPLC. In the reports of Qian et al. (2006) and Fu et al. (2015), MeOH and water were the major solvents in extraction, respectively. The solvents being used are very much in line with our sample preparations of PCRREtOH and PCRRwater. Besides, EtOH and water have been widely utilised for extraction by TCM practitioners and industries. Taken together, we believe that the reports of Qian et al. (2006) and Fu et al. (2015) can be employed as profound references/guidance in obtaining the chemical composition of PCRR. In view of your comment, we have modified the text in line 205-208, page 7.”

I do not agree with the authors' answer. There are more recent studies that show that compounds not included in this study, are observed not only in acetone extracts, but also in ethanolic extracts: 10.3390/pharmaceutics13111764.

In my opinion, this information should appear in the discussion, along with citing more recent publications: https://doi.org/10.3390/molecules24061136, DOI: 10.1055/a-0605-3857, 10.3390/pharmaceutics13111764

Author Response

Reply to the reviewer#1

Comment 1: Line 203. I do not agree with one answer of the authors.

I do not agree with the authors' answer. There are more recent studies that show that compounds not included in this study, are observed not only in acetone extracts, but also in ethanolic extracts: 10.3390/pharmaceutics13111764.

In my opinion, this information should appear in the discussion, along with citing more recent publications: https://doi.org/10.3390/molecules24061136, DOI: 10.1055/a-0605-3857, 10.3390/pharmaceutics13111764

Our response: In light of the comment from the reviewer, we have replaced an old article of Qian et al. (2006) by a recent report of Nawrot-Hadzik et al. (2021) in the Result section. Besides, we have added the related information and the other two references in Discussion section. Please see line 205-208, page 7-8 and line 270-275, page 9.